Knowledge of blood loss at delivery among postpartum patients

Farber Michaela K. mkfarber@partners.org 1
Miller Claire M. 2
Ramachandran Bharathi 2
Hegde Priya 2
Akbar Kulsum 2
Goodnough Lawrence Tim 3
Butwick Alexander J. 2
1 Department of Anesthesiology, Brigham and Women’s Hospital, Harvard Medical School , Boston , United States
2 Department of Anesthesiology, Perioperative and Pain Medicine, Stanford University School of Medicine , Stanford , CA , United States
3 Departments of Pathology and Medicine, Stanford University School of Medicine , Stanford , CA , United States
Erez Offer
Electronic publication date: 2016 Aug 31
Publication date: 2016
Volume: 4
Electronic Location ID: e2361
Received 2016 Jun 3; Accepted 2016 Jul 23
Copyright: ©2016 Farber et al.
Copyright year: 2016
Copyright holder: Farber et al.
License: This is an open access article distributed under the terms of the Creative Commons Attribution License, which permits unrestricted use, distribution, reproduction and adaptation in any medium and for any purpose provided that it is properly attributed. For attribution, the original author(s), title, publication source (PeerJ) and either DOI or URL of the article must be cited.
License URL: https://creativecommons.org/licenses/by/4.0/

Keywords: Estimated blood loss, Anemia, Postpartum hemorrhage, Patient knowledge

Funding: The authors received no funding for this work.

==============================
Background

Postpartum hemorrhage (PPH) is a leading cause of obstetric morbidity. There is limited understanding of patients’ knowledge about blood loss at delivery, PPH, and PPH-related morbidities, including transfusion and anemia.

Methods

We surveyed 100 healthy postpartum patients who underwent vaginal or cesarean delivery about blood loss, and whether they received information about transfusion and peripartum hemoglobin (Hb) testing. Responses were compared between women undergoing vaginal delivery vs. cesarean delivery; P < 0.05 considered as statistically significant.

Results

In our cohort, 49 women underwent vaginal delivery and 51 women underwent cesarean delivery. Only 29 (29%) of women provided blood loss estimates for their delivery. Women who underwent cesarean delivery were more likely to receive clear information about transfusion therapy than those undergoing vaginal delivery (43.1% vs. 20.4% respectively; P = 0.04). Women who underwent vaginal delivery were more likely to receive results of postpartum Hb tests compared to those undergoing cesarean delivery (49% vs. 29.4%; P = 0.02).

Conclusion

Our findings suggest that women are poorly informed about the magnitude of blood loss at delivery. Hematologic information given to patients varies according to mode of delivery. Further research is needed to better understand the clinical implications of patients’ knowledge gaps about PPH, transfusion and postpartum anemia.

Introduction

In the United States, the rate of severe postpartum hemorrhage (PPH) has been steadily increasing (Callaghan, Kuklina & Berg, 2010; Kramer et al., 2013). In order to decrease the frequency of PPH, clinical guidelines have been published to optimize PPH management practices (ACOG, 2006; American Society of Anesthesiologists Task Force on Perioperative Blood Management, 2015; Main et al., 2015). Obstetric and anesthetic care providers may also obtain updates about PPH management from literature review and other educational forums, such as seminars and conferences. However, it is uncertain whether patients receive information about PPH and PPH-related morbidities, such as transfusion and postpartum anemia.

If patients are inadequately informed about PPH, transfusion, and postpartum anemia, this may have important clinical and health-related implications. Firstly, PPH is recognized as an important cause of postpartum anemia. Women who develop postpartum anemia may be at risk for anemia-related morbidities, including: postpartum depression, reduced cognition, and impaired maternal-neonatal bonding (Milman, 2011). Secondly, patients who experience PPH may not receive postpartum counseling. This may negatively impact on how patients cope with the emotional trauma of experiencing major PPH (Thompson, Roberts & Ellwood, 2011). Thirdly, patient-centered care and shared decision-making about transfusion have been promoted in the perioperative and medical literature (Friedman et al., 2012; Vetter et al., 2014; Weiner et al., 2013). These approaches have not been well described in the obstetric setting, therefore examining patients’ knowledge of anticipated and actual blood loss at delivery may help inform clinical practice.

To evaluate patients’ knowledge and perceptions of postpartum blood loss, we surveyed a cohort of women who underwent vaginal delivery or cesarean delivery at a US tertiary obstetric center. We secondarily examined whether patients receive information from their care providers about transfusion, and antepartum and postpartum Hb levels.

Methods

This study was approved by Stanford University IRB, Stanford, CA (Protocol#26391). Using a convenience sample, we enrolled 100 healthy (ASA physical status 1 or 2) patients who underwent vaginal delivery or cesarean delivery at Lucile Packard Children’s Hospital, a tertiary obstetric center in California, USA. During the postpartum hospitalization, postpartum patients were approached and written informed consent was obtained. We excluded women with psychological disorders or psychiatric disease.

For this study, we asked patients two sets of questions about blood loss. One set of questions assessed patients’ baseline knowledge of normal blood loss following an uncomplicated vaginal or cesarean delivery. The second set of questions was related to the blood loss that occurred for their actual delivery (vaginal or cesarean). For each set of questions, a trained study investigator (PH, BR, KA) surveyed patients using a written questionnaire and recorded patients’ responses. Survey questions are presented in Supplemental Information 1. The questionnaire also contained questions related to patients’ socioeconomic status and educational background.

For the first set of questions, we asked patients to quantify volumes of blood loss for a normal, uncomplicated vaginal delivery and cesarean delivery. For the second set of questions, we asked patients to quantify the estimated blood loss for their actual delivery (hereafter referred to as EBLpatient), and to indicate whether an obstetric care provider informed them of their EBL. For each patient’s delivery hospitalization, we abstracted demographic, medical, obstetric and laboratory data from the electronic medical record, including: total EBL for their delivery (hereafter referred to as EBLdelivery), the antenatal hemoglobin (Hb) level most proximate to delivery, the postpartum Hb level measured closest to the day of hospital discharge, and relevant transfusion data.

For our secondary analysis, we asked directed questions related to transfusion and Hb testing. We assessed whether patients were given information during the antenatal period about transfusion, and whether they would consent to a transfusion, if clinically indicated. We asked patients whether they received information about their antenatal and postpartum Hb levels from obstetric care providers.

Statistical analyses

Data are presented as mean (standard deviation), median [interquartile range], and number (percentages), as appropriate. For continuous data, we assessed normal distributions using QQ plots and the Kolmogorov–Smirnov test. We compared patient characteristics and survey responses between women who underwent vaginal vs. cesarean delivery with a t test or Mann–Whitney test for continuous data, and χ2 test or Fisher’s exact test for categorical data. We compared EBLpatient values to EBLdelivery values for women who underwent vaginal and cesarean delivery respectively, using Wilcoxon signed rank sum test.

Using EBL data, we classified PPH using the following EBL thresholds: ≥500 ml EBL for vaginal delivery and ≥1,000 ml EBL for cesarean delivery. We calculated sensitivity, specificity, positive predictive value (PPV) and negative predictive value (NPV) to determine whether PPH was accurately classified by patients’ EBL estimates for their actual delivery. Statistical analysis was performed using STATA version 12 (Stata Corp., College Station, TX). P < 0.05 was considered as statistically significant.

Results

A total of 100 patients were recruited, of which 49 underwent vaginal delivery and 51 underwent cesarean delivery. Demographic, socioeconomic, and obstetric characteristics for the full cohort and for women stratified by mode of delivery are presented in Table 1. In the full cohort, the majority of women had private health insurance, were Caucasian or Asian, married, and had an annual household income of at least $50,000. Compared to women who underwent vaginal delivery, women who underwent cesarean delivery were older, had a higher parity, were delivered at a later gestational age, and were more likely to have undergone prior cesarean delivery.

Table 1 Maternal characteristics.

	All deliveries (n = 100)	Vaginal deliveries (n = 49)	Cesarean deliveries (n = 51)	P value	
Maternal age (y)	33 (6)	30 (5)	36 (6)	<0.001	
Race/Ethnicity:				0.54	
Caucasian	51 (51.0%)	27 (55.1%)	24 (47.1%)		
Asian	32 (32.0%)	14 (28.6%)	18 (35.3%)		
African–American	2 (2.0%)	0 (0.0%)	2 (3.9%)		
Other	15 (15.0%)	8 (16.3%)	7 (13.7%)		
Insurance type:				0.08	
Private	81 (81.0%)	36 (73.5%)	45 (88.2%)		
Public	19 (19.0%)	13 (26.5%)	6 (11.8%)		
Parity	1 [0–1]	0 [0–1]	1 [0–1]	0.03	
Highest level of education:				0.61	
Less than college	23 (23.0%)	13 (26.5%)	10 (19.6%)		
College degree	26 (26.0%)	11 (2 2.4%)	15 (29.4%)		
Graduate degree	51 (51.0%)	25 (51.0%)	26 (51.0%)		
Annual household income:				0.45	
Less than $10,000	2 (2.0%)	2 (4.1%)	0 (0.0%)		
Between $10,000–$49,000	19 (19.0%)	10 (20.4%)	9 (17.6%)		
Equal to or greater than $50,000	75 (75.0%)	35 (71.4%)	40 (78.4%)		
Missing	4 (4.0%)	2 (4.1%)	2 (3.9%)		
Marital status:				1.00	
Married	91 (91.0%)	45 (91.8%)	46 (90.2%)		
Unmarried—lives with other adults	7 (7.0%)	3 (6.1%)	4 (7.8%)		
Unmarried—lives without other adults	1 (1.0%)	0 (0.0%)	1 (2.0%)		
Unknown	1 (1.0%)	1 (2.0%)	0 (0.0%)		
Gestational age at delivery (weeks)	39 [38–39]	39 [38–40]	39 [37–39]	0.02	
Prior cesarean delivery	30 (30.0%)	2 (4.1%)a	28 (54.9%)	<0.001	
Multiple gestation:				1.00	
Singleton	97 (97.0%)	48 (98.0%)	49 (96.1%)		
Twins or higher-order	3 (3.0%)	1 (2.0%)	2 (3.9%)		
Known history of anemia or coagulation disorder	6 (6.0%)	4 (8.2%)	2 (3.9%)	0.43	
Notes.

Data presented as mean (SD), median [IQR], and n(%).

a Missing data for 1 patient.

Data related to the first set of questions about blood loss for an uncomplicated vaginal or cesarean delivery are presented in Table 2. Over two-thirds of patients did not provide estimates for normal blood loss after an uncomplicated vaginal or cesarean delivery. Among those who were willing to provide estimates, patients reported that the mean normal blood loss is higher after an uncomplicated cesarean delivery compared with an uncomplicated vaginal delivery.

The median [IQR] EBLdelivery values were significantly higher for women who underwent cesarean delivery compared to vaginal delivery (730 [600–1,000] ml vs. 250 [200–300] ml respectively; P < 0.001). A total of 18 women experienced PPH: four of these women underwent vaginal delivery, and 14 underwent cesarean delivery. Of note, no patients received transfusion.

Complete data on EBLpatient and EBLdelivery values were available for only 29 patients (Fig. 1). For those with complete data who underwent vaginal delivery (n = 16), EBLpatient values were significantly higher than EBLdelivery values (400 ml [300–578] ml) vs. 250 [200–300 ml] respectively; P = 0.02). In contrast, for those with complete data who underwent cesarean delivery (n = 13), EBLpatient values were significantly lower than EBLdelivery values (550 ml [400–800 ml] vs. 750 [600–1,000 ml]; P = 0.02). For the 29 patients with complete EBLpatient and EBLdelivery data, we calculated sensitivity, specificity, PPV and NPV to determine whether PPH was accurately classified according to EBLpatient values. The sensitivity was 60% (95% CI [14.7–94.7]), specificity was 83.3% (95% CI [62.6–95.3]), PPV was 42.9% (95% CI [9.9–81.6]), and NPV was 90.9% (95% CI [70.8–98.9]).

Figure 1 Recorded blood losses and patients’ estimates of blood loss according to mode of delivery.

EBL, estimated blood loss. Median (interquartile range) in blood loss. Horizontal line denotes median values, box borders refer to interquartile range, whiskers indicate range of values, circles indicate outliers (>1.5 times the interquartile range). The recorded blood loss was not documented in the medical records of 4 patients who underwent vaginal delivery and one patient who underwent cesarean delivery. A total of 32 patients who underwent vaginal delivery and 37 patients who underwent cesarean delivery did not provide estimates for blood loss at delivery.

Table 2 Survey of patients’ knowledge of normal blood loss for an uncomplicated vaginal and cesarean delivery.

	All deliveries (n = 100)	Vaginal deliveries (n = 49)	Cesarean deliveries (n = 51)	P value	
What is the normal blood loss after a vaginal delivery?	350 [350–500]a	350 [350–500]	350 [350–500]	0.70	
What is the normal blood loss after a CD?	750 [500–750]b	750 [350–750]	750 [500–750]	0.66	
Notes.

Data presented as median [interquartile range] and n (%).

TITLE CD cesarean delivery

a 39 patients for vaginal delivery and 34 patients for cesarean delivery did not know or chose not to answer this question.

b 44 patients for vaginal delivery and 32 patients for cesarean delivery did not know or chose not to answer this question.

Hb levels were not measured before or after delivery for 11 women and 20 women, respectively. Predelivery Hb levels were similar for those who underwent vaginal vs. cesarean delivery: 12.4 (1.4) g/dl vs. 12.3 (0.9) g/dl, respectively; P = 0.8. Similarly, no significant difference was observed in the last Hb measured before hospital discharge between women who underwent vaginal vs. cesarean delivery: 10.6 (1.1) g/dl vs. 10.4 (1.0) g/dl, respectively; P = 0.3.

Data of patients’ knowledge of transfusion and Hb levels are presented in Table 3. Women who underwent cesarean delivery were more likely to have received clear and understandable information about transfusion compared to women who had a vaginal delivery. A higher proportion of women undergoing cesarean delivery would agree to consent to transfusion compared to those undergoing vaginal delivery, however the difference in proportions was not statistically significant. With regard to Hb levels, patients who underwent vaginal delivery were more likely to have known their Hb level before delivery compared to those who underwent cesarean delivery. The proportion of patients who stated that their postpartum Hb level was measured was similar among women who underwent vaginal vs cesarean delivery (40.8% vs. 47% respectively; P = 0.74). However, among women who stated that their postpartum Hb level was measured, only 3 (7%) were given the test result.

Table 3 Survey of patients’ knowledge of transfusion and hemoglobin values.

	All deliveries (n = 100)	Vaginal deliveries (n = 49)	Cesarean deliveries (n = 51)	P value	
What was the quality of information you received about blood transfusion?				0.04	
Clear and understandable	32 (32.0%)	10 (20.4%)	22 (43.1%)		
Incompletely explained but I have a good understanding	41 (41.0%)	20 (40.8%)	21 (41.2%)		
Poorly explained and I have limited understanding	10 (10.0%)	6 (12.2%)	4 (7.8%)		
Not explained and I have no understanding	13 (13.0%)	10 (20.4%)	3 (5.9%)		
Missing	4 (4.0%)	3 (6.1%)	1 (2.0%)		
If a blood transfusion was needed, would you give consent?				0.09	
Yes	85 (85.0%)	38 (77.6%)	47 (92.2%)		
No	14 (14.0%)	10 (20.4%)	4 (7.8%)		
Missing	1 (1.0%)	1 (2.0%)	0		
Were you given any information about your Hb level before your delivery?				0.02	
Yes	39 (39.0%)	24 (49.0%)	15 (29.4%)		
No	57 (57.0%)	25 (51.0%)	32 (62.8%)		
Missing	4 (4.0%)	0 (0.0%)	4 (7.8%)		
Was your Hb level measured after delivery?				0.74	
Yes	44 (44.0%)	20 (40.8%)	24 (47.0%)		
No	33 (33.0%)	18 (36.7%)	15 (29.4%)		
Don’t know	22 (22.0%)	11 (22.4%)	11 (21.6%)		
Missing	1 (1.0%)	0	1 (2.0%)		
Notes.

Data presented as n(%).

TITLE Hb hemoglobin

Discussion

Our study provides insight into obstetric patients’ perceptions and knowledge of blood loss at delivery, transfusion, and laboratory testing for anemia. Over two-thirds of patients did not provide blood loss estimates for their delivery. Additionally, less than 50% of patients indicated that they received information about their pre- or post-delivery Hb levels. Lastly, the quality of transmitted information about transfusion and patients’ consent for transfusion varied according to mode of delivery. Based on our findings, a low proportion of women who deliver at a US tertiary obstetric center receive information about the clinical implications of peripartum blood loss, transfusion, and Hb testing before and after delivery.

It is unclear why the majority of women in our study did not provide blood loss (EBLpatient) values. We speculate that the reason is that many patients did not receive blood loss information after delivery. Those who did provide blood loss estimates for their delivery were relatively poor at correctly classifying PPH (sensitivity = 60%; PPV = 42.9%). One possible explanation for these findings is that, within this subcohort of women who gave blood loss estimates, women may not have been informed about the magnitude of their peripartum blood loss. In addition, it is also possible that some women correctly estimated their blood loss without receiving any EBL information from their obstetric care provider.

Although it is unclear whether patients who undergo uncomplicated deliveries need to be notified of their EBL or postpartum Hb levels, patients who experience PPH may benefit from receiving more detailed information about these indices. Thompson et al. (2011) reported that patients who experience PPH express interest in receiving information related to their delivery, and may benefit from counseling, psychological support, and assistance with physical recovery. Furthermore, physicians’ estimate of blood loss can often be lower than the actual volume of blood lost at delivery (Lilley et al., 2015; Toledo et al., 2007). Therefore, if blood loss is underestimated for women with PPH, then these women may develop anemia that goes undetected after delivery. To improve patient awareness of postpartum anemia, there may be benefit in providing patients with information sheets which contain advice about seeking medical review if they experience anemia-related symptoms (e.g., low mood, fatigue, poor cognition).

In our study, patients who underwent cesarean delivery were more likely to receive information about transfusion compared to those who underwent vaginal delivery. Obstetricians may be more likely to discuss the need for transfusion with patients who undergo cesarean delivery, as these women are at greater risk of PPH than those undergoing vaginal delivery (Bateman et al., 2010). Surprisingly, 20% of women who underwent vaginal delivery reported that they would not provide consent for a blood transfusion should the obstetrician deem it necessary. This finding is somewhat concerning as prompt transfusion therapy may be needed for women who experience severe PPH or postpartum anemia. Misconceptions about transfusion risk may explain why patients object to transfusion therapy. These misconceptions may be influenced by sociodemographic factors. For example, in a survey of patients’ perceptions of transfusion by Vetter et al. (2014), patients with a high school education or less expressed increased concern about the risk of allergic reaction, dyspnea, human immunodeficiency virus transmission, and medical error compared to those who attended college or graduate school. In a different survey examining patients’ beliefs about transfusion, Finucane, Slovic & Mertz (2000) observed that patients’ decision to receive transfusion may vary according to patient’s sex, race/ethnicity, and prior educational history. In light of these findings, counseling during the antenatal period may help allay the concerns and fears of patients who express a desire to avoid transfusion.

Antenatal and postpartum anemia can affect up to 52% and 24% women respectively (Milman, 2008; Milman, 2011). However, in our study, despite the majority of women having Hb levels measured before and after delivery, fewer than 50% indicated that they received any information regarding the results of these tests. Hb testing was less common for women who underwent vaginal delivery. To determine optimal screening practices, more population-based studies are needed to assess the frequency of postpartum anemia.

There are some limitations to our study. Our cohort size was relatively small, with patients recruited at a single, tertiary obstetric center. In addition, the majority of women had private insurance, were well educated, were Caucasian or Asian, and had an annual income of >$50,000. Therefore, the specific characteristics of our study population limit the generalizability of our findings. Further investigations are needed to assess knowledge and perceptions of blood loss among women from other sociodemographic backgrounds, including those without English proficiency. Our study cohort comprised healthy women who underwent uncomplicated vaginal or cesarean delivery. We did not collect information on indications for cesarean delivery or, if given, the timing of antenatal counseling. It is possible that the presence of select risk factors for PPH may influence if and when physicians inform patients about peripartum blood loss, anemia or transfusion. For example, the likelihood of antenatal counseling may be greater for women with antenatal conditions linked to severe PPH, such as placenta previa or accreta, than for women with uncomplicated pregnancies. Recall bias is a possibility as we performed our survey after delivery. Patients’ responses may have differed if our survey had been prospectively performed. Lastly, this was a convenience sample, therefore the proportion of patients who underwent cesarean delivery in our study cohort (51%) is not representative of the rate of cesarean delivery at LPCH (approximately 31%). In addition, in our study cohort, the proportion of women who experienced PPH (18%) is higher than reported in the literature (Bateman et al., 2010). As our study was exploratory in nature, further studies are needed to validate our findings using populations are more representative of a typical delivery population.

In conclusion, our findings suggest that obstetric patients receive limited information about peripartum blood loss, transfusion and peripartum Hb testing. In addition, patients’ understanding of transfusion and postpartum Hb testing may vary according to mode of delivery. Future qualitative studies are needed to examine whether better patient-provider communication improves patients’ understanding and awareness about the clinical implications of PPH, anemia, and transfusion therapy, and to examine alternative ways to disseminate relevant information to patients.

Supplemental Information

Supplemental Information 1 Raw study data

Click here for additional data file.

Supplemental Information 2 Study questionnaire

Click here for additional data file.

The authors would like to thank Flavya Esteves who assisted with data collection for this study.

Additional Information and Declarations

Competing Interests

Author Contributions

Human Ethics

Data Availability

The authors declare there are no competing interests.

Michaela K. Farber wrote the paper, reviewed drafts of the paper.

Claire M. Miller analyzed the data, wrote the paper, prepared figures and/or tables, reviewed drafts of the paper.

Bharathi Ramachandran, Priya Hegde and Kulsum Akbar performed the experiments, reviewed drafts of the paper.

Lawrence Tim Goodnough reviewed drafts of the paper.

Alexander J. Butwick conceived and designed the experiments, performed the experiments, analyzed the data, wrote the paper, prepared figures and/or tables, reviewed drafts of the paper.

The following information was supplied relating to ethical approvals (i.e., approving body and any reference numbers):

Stanford University IRB protocol 26391.

The following information was supplied regarding data availability:

The raw data has been supplied as Supplemental File.

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
