# Peer review of "Knowledge of blood loss at delivery among postpartum patients"

_PeerJ, doi:10.7717/peerj.2361_

## Round 0.1 · original submission · Major Revisions

· Academic Editor

Major Revisions

This manuscript was reviewed by two experts in the field and the authors need to address thier comments prior for reconsidering this manuscript for publication

Reviewer 1 ·

Basic reporting

The objective of this article is to examine patients' knowledge regarding blood loss during delivery. More specifically, the authors set to find out: (1) whether women who delivered recently were aware of the average amount of blood loss during vaginal and cesarean section delivery, (2) the amount of estimated blood loss in their own delivery, and (3) whether they were given information about blood transfusion and about their Hb level. The methodological tool used in this prospective study is questionnaires, and study population included two groups: vaginal delivery group and cesarean section group. The results demonstrate significant differences between the two study groups in regard to receiving information about transfusion and Hb testing. The authors conclude that women are not adequately informed regarding peripartum blood loss.
-The article is well structured, wording is clear, and language is coherent.
-Line 190 - may not adequately informed -> may not be adequately informed

Experimental design

Two sets of questionnaires were used. The first of which dealt with the patients’ basic knowledge about obstetric blood loss, and the second with the data the patients received regarding blood loss in the course of their delivery.

-The statistical analysis is satisfactory and well explained, enabling replication.

Validity of the findings

•The authors state that (line 190):" Those who gave blood loss estimates for their delivery were relatively poor at correctly classifying PPH (sensitivity=60%; PPV=42.9%). These findings suggest that women may not adequately informed about the magnitude of their peri-partum blood loss." This conclusion is based on answers from 29 women, and I am not sure there is sufficient evidence to the notion that "women are not adequately informed" on the basis they could not provide blood loss estimates. Does correctly guessing the amount of blood you loss during your delivery really corresponds to whether you were informed or not? This needs to be addressed in the discussion.

•The authors state that (line 204): "Excluding religious objections, explanations for why patients would object to receiving blood transfusions are not known." Reasons for refusal of blood transfusions, in the general population, have been previously reviewed in the literature. See, for example (Finucane, Slovic, & Mertz, 2000).

•Regarding (line 240) "Future studies are needed to determine whether better patient education improves patients’ understanding and awareness of PPH and PPH-related morbidity". I would be more specific here, and say that future qualitative studies are required in order to really understand patients conceptions and comprehensions in this matter, and in order to identify more efficient ways of providing information to patients.

·

Basic reporting

Clear English. Sufficient introduction and background.
Appropriate tables . Coherent .

Experimental design

Original primary research within the scope of the journal.
The research question is clear .
In the Methods there is no information regarding the indication for CS wether elective or emergent that may affect the EBL but also the counseling time regarding blood transfusions. Important to mention and emphasize .

Validity of the findings

See the above regarding CS indications as well more description of the pph etiology of both CS and vaginal delivery. I believe that whenever there is expected complication there is better counseling i.e TOLAC after 2 CS vs NVS primipara no risk factors. These may all affect the results and the authors should be a ware/ control these factors.
Table 1: There different numbers of Multiple gestations among the 2 groups w/o p value calculation. Also anemia and coagulation disorder are included in the table while on M&M were reported as excluded (at least the bleeding disorders)

Additional comments

This manuscript is important in increasing the the physician's awareness regarding counseling the patient about PPH and blood transfusions. It is extremely important to provide information to the patient particularly before and after PPH , however I am not convinced that it is important /relevant for the low risk patient with normal EBL to knows the obstetrics values of EBL and their own Hb value before or after delivery.(particularly as those are only rough estimates)
These values require knowledge of the normal range of Hb and distribution of the normal range male/ female and the physiology of pregnancy. It can be overwhelming and confusing, It is though very important to talk about anemia in general and particularly in those cases of PPH where blood transfusion were given .In these cases it is acceptable to mention the EBL and perhaps by providing more information as Hb values may be useful, but definitely not as standard of care.
I recommend on focusing on that a aspect in the information presentation and discussion as well to increase the study sample and specify the indications for cs /pph in order to make this manuscript more concise.

---

## Round 0.2 · accepted · Accept

· Academic Editor

Accept

The authors have well adressed the reviewers comments and the manuscript is currently sutible for publication